# Implementation of oral health evidence-based practices in early care education settings across the U.S. during different COVID-19 periods

Phoebe P. Tchoua👁*, Shreena Patel👁, Aviva Shira Starr‡, Richard Rairigh👁‡, Falon Smith‡, Erik A. Willis👁

Center for Health Promotion and Disease Prevention, The University of North Carolina at Chapel Hill, Chapel Hill, North Carolina, United States of America

👁 These authors contributed equally to this work.
‡ These authors also contributed equally to this work.
* ptc@unc.edu (PPT)

## Abstract

The COVID-19 pandemic disrupted oral health practices in early care education (ECE) centers. This study describes the implementation of oral health evidence-based practices (EBP) in ECE centers enrolled in the web-based Go NAPSACC program pre-, during-, and post-COVID-19 stay-at-home (SAH) orders. This repeated cross-sectional study analyzed retroactive data from three types of programs (n = 1,490), that participated in Go NAPSACC oral health modules between January 2017 and April 2024: Head Start (n = 154), family child care home (FCCH; n = 540), and center-based (n = 796). Programs that did not use the Oral Health module (n = 10,425) and had duplicate registrations (n = 91) were excluded. The analysis focused on EBP total score and percentage of EBP met scores. We found significant differences in oral health EBP total and EBP met scores between program type (p < 0.001). Head Start programs had statistically significant higher EBP total percentage scores (81.8, 95% confidence interval [CI] = 78.5, 85.2; p < 0.0001) than FCCH programs (69.5, 95% CI = 67.1, 71.8; p < 0.0001), and center-based programs (59.5, 95% CI = 57.3, 61.7). Similarly, Head Start programs had higher EBP met scores (62.0, 95% CI = 57.7, 66.3; p < 0.0001), than FCCH programs (49.7, 95% CI = 46.7, 52.7; p < 0.0001), and center-based programs (36.9, 95% CI = 34.1, 39.8). We observed no statistically significant differences among programs based on SAH order period for neither EBP total scores (period, p = 0.761; interaction between program type and period, p = 0.788) nor EBP met scores (period, p = 0.178; interaction between program type and SAH order period, p = 0.293). These findings suggest that ECE programs struggle to meet oral health EBP across the three study periods, and the observed differences across program type was not explained by SAH orders.

**Data availability statement:** Due to contractual agreements with certain state partners, Go NAPSACC data cannot be made publicly available. However, de-identified or limited datasets may be made available upon reasonable request, contingent upon review and approval by the project team and compliance with institutional and contractual requirements. Contact: Go NAPSACC Initiative Team, gonapsacc@unc. edu.

**Funding:** First author, Dr. Tchoua, is supported by the T32 Cancer Health Disparities Training Grant (T32CA128582) from the National Cancer Institute of the National Institutes of Health. Service agreement contracts with state departments, non-profit, and academic institutions support Go NAPSACC's dissemination, allowing for the collection of data reported in this study. The funders had no role in study design, data collection and analysis, decision to publish, or preparation of the manuscript.

**Competing interests:** The authors have declared that no competing interests exist.

## Introduction

Early childhood caries (ECC) is the most common chronic disease among American children, affecting 21.4% of children 2–5 years old and poses a significant risk to their general health [1,2]. According to the American Academy of Pediatric Dentistry, ECC is classified as the "the presence of 1 or more decayed (non-cavitated or cavitated lesions), missing (due to caries), or filled tooth surfaces in any primary tooth" in children under six years of age [3]. Most ECC are preventable through health promotion practices that focus on foods provided, oral hygiene, fluoride use, access to dental care, caregiver education, and feeding practices [4]. Developing and encouraging proper oral care routines in infancy and early childhood sets the groundwork for long-term oral health [2].

In 2019, 62% of American children who received weekly non-parental care were enrolled in out-of-home early care and education (ECE) programs [5], with children under age five spending an average of 24 hours per week in these settings [6]. During this time, children often consume a substantial portion of their daily nutrition – up to two meals and two snacks [7]. Given the substantial role ECE programs play in children's daily routines, they are uniquely positioned to promote oral health and prevent ECC. By implementing effective organizational policies, practices, and oral health promotion strategies, ECE settings can help establish healthy habits that support long-term oral health for all children in their care [8].

A review of state regulations revealed significant gaps in oral health policies across six key categories: screening for dental care needs, referral for dental care, storing toothbrushes, tooth brushing, fluoride use, and oral health education. Fifteen states lacked any regulations, 23 states had only one, and 13 states had more than one [9]. Additionally, oral health practices vary between state-funded and non-state-funded ECE programs [10]. Non-state-funded programs reported higher rates of oral health education (55% vs 39%) and oral health practices (85% vs 68%), whereas state-funded programs were more likely to include routine toothbrushing (46% vs 21%) [2,11]. These discrepancies in policies and practices led the American Academy of Pediatric Dentistry to issue a policy statement urging ECE settings to adopt standardized oral health guidelines. Their recommendations aim to minimize the risk of ECC by emphasizing dental disease prevention and oral health promotion [11].

The COVID-19 pandemic may have further disrupted oral health practices in ECE settings [12]. In 2020, parents were 16% less likely to perceive their children's dental health as excellent compared to 2019 [13], and reports of poor dental health increased by 75% [13]. ECE programs faced significant challenges, including temporary and permanent program closures and decreased enrollment, that hindered their ability to operate effectively [14]. These disruptions impacted not only the sustainability of programs but also the continuity of care and education for young children. The decline in oral health and the operational challenges faced by ECE programs are closely interconnected [15,16]. To mitigate the risk of COVID-19 transmission, many routine oral health practices in ECE settings were either halted or significantly modified. For example, a COVID-19 Head Start policy suspended tooth brushing in

classrooms – a practice previously required in many programs [12]. This action, though crucial infection control, may have inadvertently impacted the oral hygiene habits of young children.

This study aims to address critical gaps in understanding how the implementation of oral health practices in ECE settings have evolved over time, particularly in response to the disruptions caused by the COVID-19 pandemic. Given the significant role ECE programs play in supporting young children's health and the notable variability in oral health policies and practices across states and program types, it is essential to evaluate how these settings adapted to pandemic-related challenges. Understanding the extent to which oral health practices were maintained, modified, or discontinued during key stages of the pandemic can inform future strategies to promote oral health in ECE environments. Moreover, insights gained from this study will help identify areas where additional support, training, or policy changes are needed to ensure that ECE programs are equipped to sustain effective oral health practices. This work is particularly relevant for guiding the continued implementation of evidence-based health promotion programs to improve child health outcomes.

## Methods

The data for this repeated cross-sectional study are from ECE programs enrolled in the web-based Go NAPSCC program (gonapsacc.org) from January 2017 to April 2024 and was conducted on September 3, 2024. Go NAPSACC, an evidence-based health promotion initiative for ECE settings, helps programs implement evidence-based practices across seven health modules, including oral health. There are 12,048 ECE programs registered with Go NAPSACC across 23 states [15,16]. ECE programs identified as duplicates (n = 91) and those not participating in the Oral Health module (n = 10,425) were excluded. Due to low enrollment of school-based ECE programs (n = 42), we limited this analysis to only those that were classified as Head Start (n = 154), family child care homes (FCCH; n = 540), or center-based (n = 796) resulting in an analysis cohort of 1,490 ECE programs. Using a repeated cross-sectional design, self-assessments completed during three time periods corresponding to key times of the COVID-19 epidemic in United States: (1) pre-stay at home (SAH) order; (2) during SAH order; (3) post-SAH were included in the analysis. This study analyzed available organizational level data with no individual level identifiable personal information, therefore ethics approval and informed consent were not required.

### Program characteristics

Program data were self-reported by ECE administrators through the Go NAPSACC online system. Administrators report program type (e.g., Head Start, center-based, FCCH), association (e.g., faith-based, military), Child and Adult Care Food Program (CACFP) participation, care type (full-/half-day), ages served, provided meals, subsidies received, years in operation, and number of children enrolled.

### Oral health evidence-based practices

Evidence-based practice recommendations around oral health were created through extensive reviews of the scientific literature, authoritative recommendations (e.g., Office of Head Start, American Dental Association, American Academy of Pediatrics Dentistry (AAPD), Caring for our Children, and Oral Health expert opinion). These recommendations were used to create a self-assessment tool completed by ECE administrators and are scored using a 4-point Likert-type scale, from 1 = "not engaging", 2 = "minimally engaging", 3 = "somewhat engaging", to 4 = "fully engaging" in evidence-based practice recommendations.

The oral health module consisted of 25 questions under five evidence-based practice domains: tooth brushing (six questions), food and beverages provided (five questions), teacher practices (six questions), education and professional development (seven questions), and policy (one question). Tooth brushing domain asked questions on time allocated for toothbrushing per week, how the toothpaste was distributed, toothpaste availability, and the frequency of use of fluoride

toothpaste. Food and beverages domain asked questions on how often high sugar foods, flavored milk, 100% juice, and regular juice were offered (i.e., daily, weekly, or monthly), and how water was made available (i.e., upon request, freely available, visibly accessible, offered indoors or outdoors). Teacher practices domain asked questions on whether bottles or sippy cups were offered to infants and toddlers (i.e., at naptime or playtime), the level of tooth brushing assistance teachers provided to children, and whether teachers offered praise and a positive environment during tooth brushing sessions. Education and professional development domain asked questions on how often teachers used planned oral health education in their lessons, received professional development, provided families information on the child's oral health throughout the year, and spoke to children about the importance of oral health indirectly. Policy domain asked a question to assess the ECE program's written policies on oral health.

### Evidence-based practice score

Evidence-based practice (EBP) total scores are calculated by summing all scored items in the self-assessment divided by the total possible points for all applicable items and the total is multiplied by 100, yielding a percentage score between 0 (least engaging) to 100 (most engaging).

### Percentage of evidence-based practices met

EBP met percentage score is calculated by summing the number of items in the self-assessment where the best practice was fully engaged (score = 4) and dividing this number by the total number of evidence-based practices and the total is multiplied by 100, yielding a percentage score between 0 (not fully engaging) to 100 (fully engaging). The EBP total score and EBP met percentage score helps measure the full range of ECE program's engagement in evidence-based practices.

### SAH order data source

Retrospective data was obtained from individual state-maintained websites to identify the exact dates when SAH orders were enacted for each state. Data collected prior to this date were classified as pre-SAH order period. Data collected between the date of the SAH order (between March 4, 2020, and April 7, 2020) and August 4, 2021 (the date when the Office of Head Start lifted its toothbrushing suspension in ECE programs), was classified as during the SAH order period. Any data collected after August 4, 2021, was classified as post-SAH order period.

### Statistical analyses

Data were summarized using means and standard deviations for continuous variables, and frequencies and percentages for categorical variables. If a program had more than one oral health self-assessment, their first completed self-assessment was used in the analysis. Generalized linear mixed models (SAS PROC MIXED) were used to compare evidence-based practice and percent of evidence-based practices met scores between program type (i.e., Head Start, FCCH, center-based) across SAH periods. Models included fixed effects for program type, SAH order period (pre, during, post), program type*SAH interaction, and Go NAPSACC registration date. Random effect for the state the ECE program resided was included to account for correlations among programs operating within in the same state. Models included a subject-specific random effect and accounted for the clustering of ECE programs within the same state. Statistical significance was determined at 0.05 alpha level, and all analyses were performed in spring 2024 using SAS version 9.4 (SAS Institute Inc., Cary, NC).

### Results

Most Go NAPSACC programs were center-based (n = 796, 53.4%) with no program association (n = 1,282, 86.0%). They were predominantly located in metropolitan areas (n = 1,032, 69.3%), participated in CACFP (n = 1,025, 68.8%), and

operated on a full-day schedule (n = 1,434, 96.2%). These programs served children aged 0–5 years and had been in operation for an average of 19.2 years (SD = 15.4). See Table 1

## Oral health module during SAH periods

ECE programs were impacted in various ways pre-, during-, and post-SAH periods. During the SAH period, the number of Head Start programs initiating the oral health module decreased (14.8% to 13.4%) and remained low post-SAH (4.9%). Conversely, the number of FCCH programs initiating work on oral health increased (28.7% to 34.9%) and remained high (43.5%). Center-based programs experienced less fluctuation (56.6% to 51.7% to 51.6%). The location of ECE programs also played a factor in those choosing to work on oral health promotion. While work on the oral health module in rural programs increased (2.0% to 3.1% to 5.3%), urban environments experienced an initial decrease (29.0% to 24.9%) that rebounded post-SAH (26.6%).

## Evidence-based practice total score

Mean EBP total percentage scores by program type and SAH order period are presented in Fig 1. Across all three periods, Head Start programs consistently had the highest EBP total percentage score (82.37%, 82.34%, 80.83%), followed by FCCH programs (70.0%, 69.74%, 68.67%), and lastly center-based programs (59.96%, 58.62%, 59.86%). Further analysis showed a significant difference in EBP scores between program type (p < 0.001). Specifically, center-based

**Table 1. Participant Characteristics by three key periods of the COVID-19 Pandemic.**

|  | All Participants | PRE-SAH | DURING-SAH | POST-SAH |
|---|---|---|---|---|
|  | n (%) | n (%) | n (%) | n (%) |
| **Type of program** |  |  |  |  |
| Head Start | 154 (10.3%) | 80 (14.8%) | 43 (13.4%) | 31 (4.9%) |
| Family Child Care Home | 540 (36.2%) | 155 (28.7%) | 112 (34.9%) | 273 (43.5%) |
| Center-based | 796 (53.4%) | 306 (56.6%) | 166 (51.7%) | 324 (51.6%) |
| **Program association** |  |  |  |  |
| Faith-based | 21 (1.4%) | 20 (3.7%) | 1 (0.3%) | 0 (0.0%) |
| Native American/Alaska Native Tribe | 8 (0.5%) | 4 (0.7%) | 0 (0.0%) | 4 (0.6%) |
| Military | 17 (1.1%) | 8 (1.5%) | 5 (1.6%) | 4 (0.6%) |
| Multiple | 162 (10.9%) | 47 (8.7%) | 40 (12.5%) | 75 (11.9%) |
| None | 1282 (86.0%) | 462 (85.4%) | 275 (85.7%) | 545 (86.8%) |
| **Urbanization** |  |  |  |  |
| Rural | 54 (3.6%) | 11 (2.0%) | 10 (3.1%) | 33 (5.3%) |
| Urban | 404 (27.1%) | 157 (29.0%) | 80 (24.9%) | 167 (26.6%) |
| Metropolitan | 1032 (69.3%) | 373 (68.9%) | 231 (72.0%) | 428 (68.2%) |
| **Participates in CACFP** | 1025 (68.8%) | 386 (71.3%) | 228 (71.0%) | 411 (65.4%) |
| **Enrollment type** |  |  |  |  |
| Full-day | 1434 (96.2%) | 522 (96.5%) | 306 (95.3%) | 606 (96.5%) |
| Half-day | 56 (3.8%) | 19 (3.5%) | 15 (4.7%) | 22 (3.5%) |
| **Ages served** |  |  |  |  |
| 0 to 2 years (yes) | 1284 (86.2%) | 456 (84.3%) | 269 (83.8%) | 559 (89.0%) |
| 2 to 5 years (yes) | 1460 (98.0%) | 530 (98.0%) | 318 (99.1%) | 612 (97.5%) |
| Years in operation (mean [SD]) | 19.2 (15.4) | 21.8 (15.6) | 20.0 (14.9) | 16.5 (15.2) |
| Total number of children | 71,399 | 29,927 | 13,801 | 27,671 |

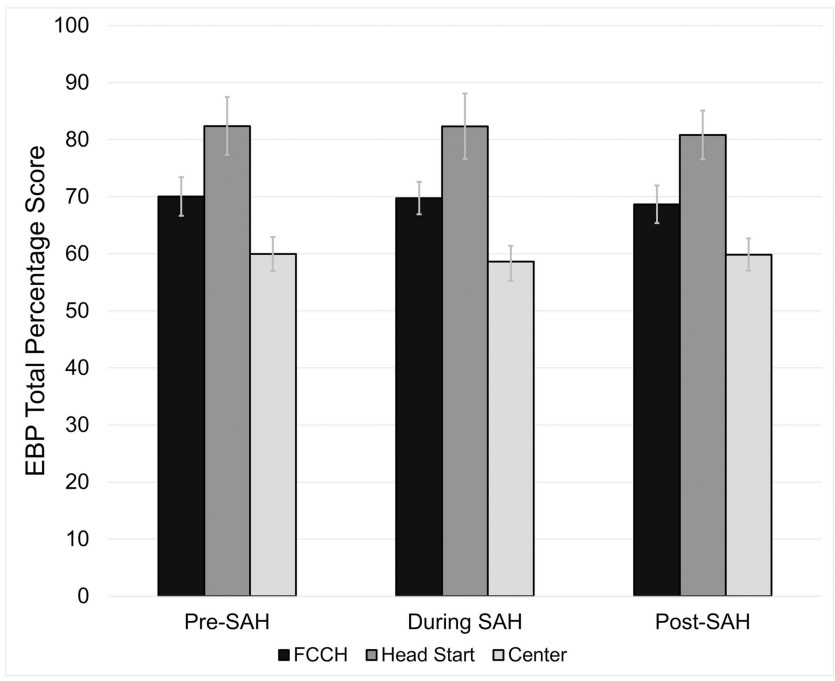

**Fig 1. Oral health evidence-based practice scores across different early childhood education program types during three key periods of the covid-19 pandemic.**

programs had significantly lower scores (59.5, 95% confidence interval [95% CI] = 57.3, 61.7) than both Head Start (81.8, 95% CI = 78.5, 85.2; <0.0001) and FCCH programs (69.5, 95% CI = 67.1, 71.8; <0.0001). However, there were no statistically significant differences observed among programs based on SAH order period (period, p = 0.761; interaction between program type and period, p = 0.788).

### Evidence-based practice met percentage score

No program fully met the EBP recommendations (i.e., scored 100%). The EBP met percentage scores by program type and SAH order period are presented in Fig 2. Like EBP total percentage score, Head Start programs had the highest EBP met percentage scores (64.38%, 64.80%, 56.95%), followed by FCCH programs (50.93%, 49.68%, 48.50%), and center-based programs (37.01%, 36.65%, 37.18%). Statistical analysis showed a significant difference between program type (p < 0.001). Specifically, center-based programs had significantly lower scores (36.9, 95% CI = 34.1, 39.8) than both Head Start (62.0, 95% CI = 57.7, 66.3; <0.0001) and FCCH programs (49.7, 95% CI = 46.7, 52.7; <0.0001). However, there were no statistically significant differences observed among programs based on the SAH order period (period, p = 0.178; interaction between program type and SAH order period, p = 0.293).

### Discussion

This study assessed the implementation of ECE oral health EBP during three COVID-19 time periods (i.e., pre-SAH, during-SAH, and post-SAH). Our findings revealed that Head Start programs consistently had higher EBP total and EBP met percentage scores than FCCH and center-based programs. Head Start program administrators' self-assessment results reveal that their programs were the most engaged in oral health evidence-based practice recommendations than the other two program types. Based on our findings, government issued SAH orders did not explain the difference in EBP scores observed across the three programs.

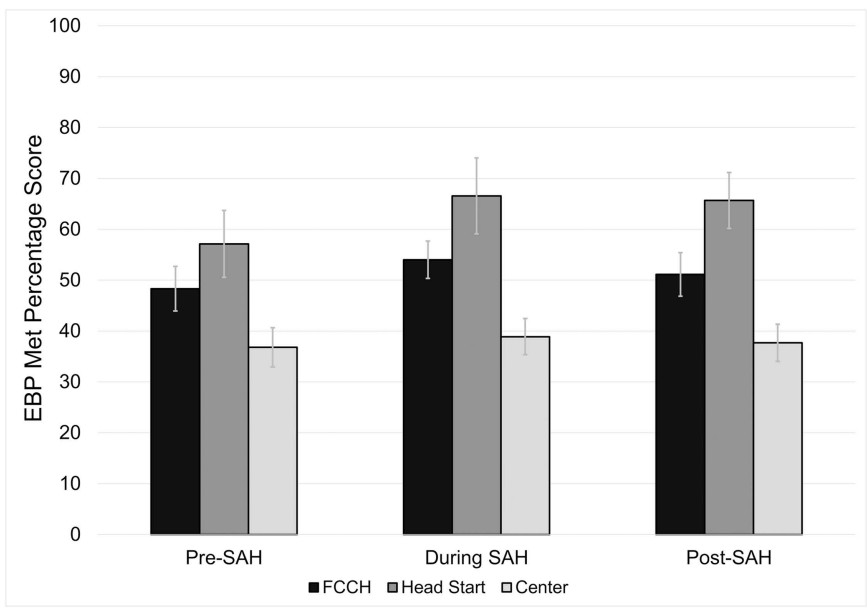

**Fig 2.** Oral health evidence-based met percentage scores across different early childhood education program types during three key periods of the covid-19 pandemic.

The primary cause for the difference in EBP total and EBP met percentage scores among all three programs is unclear. Perhaps, funding structure, although complex, can help explain the differences in scores we found. Centers' oral health practices can differ based on funding source, state-funded, non-state funded [10]. Scheunemann et al. (2015) compared toothbrushing as a routine classroom activity and funding source in ECE programs across the state of Wisconsin. They found that state-funded ECE centers were over two times more likely to have this practice than non-state funded. However, non-state funded were five times more likely to have any oral health educational practice. Head Start programs are federally funded and comprehensive and offer multiple services to children 0–5 years old and their families at no cost. These services include early learning and development, health and wellness, family well-being, and family engagement. Under health and wellness, oral health practices are promoted [17]. This integration of oral health practices support could help explain why Head Start programs, with primarily federal funding, scored significantly higher than FCCH and center-based programs, which are primarily funded via families' fees.

Unlike Head Start programs, FCCH and center-based programs generally do not offer such comprehensive services and may instead focus on other areas of early care and development [18,19]. As a result, these programs may view oral health practices primarily as the responsibility of families and prioritize other efforts (e.g., child education, administrative duties, nutritional services) that align more closely with their organizational scope or resources [20]. Some ECE staff have noted barriers to implementing toothbrushing programs, including lack of time, limited educational health training, and insufficient administrative support, and increased workload [20]. However, oral health practices should be prioritized by both ECE providers and families, as recommended by the AAPD, to reduce the risk of ECC in young children.

ECE may provide the only opportunity for some children to receive oral health care education. However, ECE programs are faced with different barriers that can impede their ability to provide adequate and quality oral health education to children in their care. In a study by Joshi et al [21], ECE center directors reported funding issues and lack of training as the main self-perceived barriers to implementing oral health promotion practices in their center. Surprisingly, concerns such as time constraints, infection control, and lack of space were less common. Similarly, Joufi et al [22] found that time was the

least anticipated barrier in implementing oral health promotion practices. Lack of training was also identified as a barrier by ECE teacher in a systematic review by Joufi et al [22], as well as by ECE directors in a study by Joshi et al [23]. In addition, teachers cited inadequate resources as barriers to performing oral health activities with children [23]. However, ECE staff members found an oral health training program effective in improving the oral health of children in their centers [22]. Therefore, FCCH and center-based programs administrators, oral health leaders, and health departments who support ECE programs could focus on implementing oral health education training for program staff and provide the needed resources.

To our knowledge, this study is the first to examine how three types of ECE programs in the US implemented oral health evidence-based practices during three COVID-19 time periods. However, like most studies, it is subject to at least two limitations. First, reporting bias and social desirability bias, the data shared here is from self-assessments completed by ECE administrators. Second, this is cross-sectional study and therefore any causal link between the program type and the EBP total and EBP met scores cannot be assessed. Despite these limitations, this study adds to the literature and provides information on the implementation of oral health evidence-based practices across three types of ECE programs in 23 states, Head Start, FCCH, and center-based. These findings can inform ECE programs and policy makers in promoting oral health education trainings for teachers and administrators, as well as providing relevant resources to encourage staff and parents to consistently reinforce oral health practices with the children in their care. More broadly, further research is needed to better understand why center-based programs implement fewer oral health EBP and to identify the support needed to help them increase their EBP total and EBP met scores.

In conclusion, this study found that no child care program fully engaged in oral health evidence-based practices and Head Start programs had the highest level of engagement than FCCH and center-based programs across three COVID-19 SAH periods. SAH orders did not explain the differences in scores we observed. ECE program policies should promote ECE administrators to prioritize oral health promotion practices to prevent early childhood caries among the children they serve.

## Author contributions

**Conceptualization:** Shreena Patel, Erik A. Willis.

**Data curation:** Erik A. Willis.

**Formal analysis:** Erik A. Willis.

**Methodology:** Erik A. Willis.

**Supervision:** Phoebe Tchoua, Erik A. Willis.

**Writing – original draft:** Phoebe Tchoua, Shreena Patel.

**Writing – review & editing:** Phoebe Tchoua, Shreena Patel, Aviva Shira Starr, Richard Rairigh, Falon Smith, Erik A. Willis.

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
