## [Decision Letter · Decision Letter 0]

4 Jun 2025

Dear Dr. Tchoua,

Thank you for submitting your manuscript to PLOS ONE. After careful consideration, we feel that it has merit but does not fully meet PLOS ONE’s publication criteria as it currently stands. Therefore, we invite you to submit a revised version of the manuscript that addresses the points raised during the review process.

We look forward to receiving your revised manuscript.

Kind regards,

Nour Ammar

Academic Editor

PLOS ONE

 [This research was supported by a grant from the T32 Cancer Health Disparities Training Grant from the National Cancer Institute of the National Institutes of Health (T32CA128582).]. 

Reviewers' comments:

Reviewer's Responses to Questions

**Comments to the Author**

1. Is the manuscript technically sound, and do the data support the conclusions?

Reviewer #1: Yes

Reviewer #2: Yes

2. Has the statistical analysis been performed appropriately and rigorously?

Reviewer #1: Yes

Reviewer #2: Yes

3. Have the authors made all data underlying the findings in their manuscript fully available?

Reviewer #1: Yes

Reviewer #2: Yes

4. Is the manuscript presented in an intelligible fashion and written in standard English?

Reviewer #1: Yes

Reviewer #2: Yes

Reviewer #1: I appreciate the opportunity to review this manuscript. The topic is relevant and addresses an issue with potential impact in the field of public health. The manuscript is generally well-structured, but there are areas that could be improved for greater clarity and scientific rigor.

1) Abstract: The methods section is very brief. It does not mention that this is a cross-sectional study (which is an essential item according to the Strengthening the Reporting of Observational Studies in Epidemiology [STROBE] statement), nor does it specify that the data used span the period between July 2014 and February 2024. Additionally, the exclusion criteria were not described.

2) Introduction: The introduction is well written; however, it lacks a description of the specific objectives of the study, including any pre-existing hypotheses.

3) Methods: Was any measure taken to reduce or control risk of bias? If so, please specify.

Reviewer #2: 1. Some refrences are either missing or are bit old.

2. Certain typographical errors

3. some inccomplete sentences or imbigious

**Do you want your identity to be public for this peer review?** For information about this choice, including consent withdrawal, please see our Privacy Policy

Reviewer #1: **Yes: ** Jennifer Reis-Oliveira

Reviewer #2: **Yes: ** Dr. Faiza Awais

---

## [Author Response · Author response to Decision Letter 1]

10 Jul 2025

Reviewer #1: 

Comment 1. Abstract: The methods section is very brief. It does not mention that this is a cross-sectional study (which is an essential item according to the Strengthening the Reporting of Observational Studies in Epidemiology [STROBE] statement), nor does it specify that the data used span the period between July 2014 and February 2024. Additionally, the exclusion criteria were not described.

Response 1. Thank you for your comment. We revised the abstract to specify that the study is repeated cross-sectional, covers the period from July 2014 to February 2024, and includes the exclusion criteria. See Lines 26, 27-28, and 29-30.

Comment 2. Introduction: The introduction is well written; however, it lacks a description of the specific objectives of the study, including any pre-existing hypotheses.

Response 2. Thank you for your comment. This study is descriptive and does not have a pre-existing hypothesis. We clarified the objective as understanding the implementation of oral health practices in early care education settings before, during, and after the COVID-19 pandemic. See lines 104-106

Comment 1. Methods: Was any measure taken to reduce or control risk of bias? If so, please specify.

Response 1. Thank you for your comment. Yes. To minimize bias, (1) we defined exclusion criteria for data analysis (i.e., programs who did not participate in the Oral Health module), (2) our data was collected using a uniform self-assessment across all program participants, and (3) we acknowledged our limitations, self-reported data, and study design.

Reviewer #2:

Comment 1. Some references are either missing or are bit old.

Response 1. Thank you for your comment. We updated the manuscript with the most recent available references related to children’s oral health. Unfortunately, we are limited by currently available references.

Comment 2. Certain typographical errors

Response 2. We appreciate the reviewer bringing these to our attention. We have addressed typographical errors in the revised manuscript.

Comment 3. Some incomplete sentences or ambiguous

Response 3. Thank you for your comment. We carefully reviewed and corrected typographical errors throughout the manuscript.

Manuscript Edits

Comment 1. Introduction. “The COVID-19 pandemic disrupted many health practices across the US, including oral health practices in ECE settings. (reference).”

Response 1. We revised the sentence and added a reference 12. See lines 90-91.

Comment 2. Introduction. “These disruptions impacted not only the sustainability of programs but also the continuity of care and education for young children. The decline in oral health and the operational challenges faced by ECE programs are closely interconnected.(reference).”

Response 2. We added references 15 and 16 to support this statement. See lines 95-98.

Comment 3. Methods. “There are 11,431 ECE programs registered with Go NAPSACC across 23 states.(reference).”

Response 3. We added references 15 and 16 to support this statement. See lines 123-124.

---

## [Decision Letter · Decision Letter 1]

30 Jul 2025

Implementation of oral health evidence-based practices in early care education settings across the U.S. during different COVID-19 periods

PONE-D-25-17567R1

Dear Dr. Tchoua,

We’re pleased to inform you that your manuscript has been judged scientifically suitable for publication and will be formally accepted for publication once it meets all outstanding technical requirements.

Kind regards,

Nour Ammar

Academic Editor

PLOS ONE

Additional Editor Comments (optional):

Reviewers' comments:

Reviewer's Responses to Questions

**Comments to the Author**

Reviewer #1: All comments have been addressed

2. Is the manuscript technically sound, and do the data support the conclusions?

Reviewer #1: Yes

3. Has the statistical analysis been performed appropriately and rigorously?

Reviewer #1: Yes

4. Have the authors made all data underlying the findings in their manuscript fully available?

Reviewer #1: Yes

5. Is the manuscript presented in an intelligible fashion and written in standard English?

Reviewer #1: Yes

Reviewer #1: (No Response)

**Do you want your identity to be public for this peer review?** For information about this choice, including consent withdrawal, please see our Privacy Policy

Reviewer #1: **Yes: ** Jennifer Reis-Oliveira

---

## [Editor Report · Acceptance letter]

PONE-D-25-17567R1

PLOS ONE

Dear Dr. Tchoua,

I'm pleased to inform you that your manuscript has been deemed suitable for publication in PLOS ONE. Congratulations! Your manuscript is now being handed over to our production team.

Kind regards,

on behalf of

Dr. Nour Ammar

Academic Editor

PLOS ONE